# Suicide before and during the COVID-19 Pandemic: A Systematic Review with Meta-Analysis

**DOI:** 10.3390/ijerph20043346

**Published:** 2023-02-14

**Authors:** Yifei Yan, Jianhua Hou, Qing Li, Nancy Xiaonan Yu

**Affiliations:** 1Department of Social and Behavioural Sciences, City University of Hong Kong, Tat Chee Avenue, Kowloon, Hong Kong SAR, China; 2Department of Computing, The Hong Kong Polytechnic University, Hong Kong SAR, China

**Keywords:** suicidal ideation, suicide attempt, death by suicide, COVID-19 pandemic, meta-analysis

## Abstract

Synthesizing evidence to examine changes in suicide-related outcomes before and during the pandemic can inform suicide management during the COVID-19 crisis. We searched 13 databases as of December 2022 for studies reporting both the pre- and peri-pandemic prevalence of suicidal ideation, suicide attempts, or rate of death by suicide. A random-effects model was used to pool the ratio of peri- and pre-pandemic prevalence of suicidal ideation and attempt (Prevalence Ratio—PR) and rate of death by suicide (Rate Ratio; RR). We identified 51, 55, and 25 samples for suicidal ideation, attempt, and death by suicide. The prevalence of suicidal ideation increased significantly among non-clinical (PR = 1.142; 95% CI: 1.018–1.282; *p* = 0.024; *k* = 28) and clinical (PR = 1.134; 95% CI: 1.048–1.227; *p* = 0.002; *k* = 23) samples, and pooled estimates differed by population and study design. Suicide attempts were more prevalent during the pandemic among non-clinical (PR = 1.14; 95% CI: 1.053–1.233; *p* = 0.001; *k* = 30) and clinical (PR = 1.32; 95% CI: 1.17–1.489; *p* = 0.000; *k* = 25) participants. The pooled RR for death by suicide was 0.923 (95% CI: 0.84–1.01; *p* = 0.092; *k* = 25), indicating a nonsignificant downward trend. An upward trend of suicidal ideation and suicide attempts was observed during the COVID-19 pandemic, despite suicide rate remaining stable. Our findings suggest that timely prevention and intervention programs are highly needed for non-clinical adult population and clinical patients. Monitoring the real-time and long-run suicide risk as the pandemic evolves is warranted.

## 1. Introduction

Suicide constitutes a serious public health issue. Humans are usually vulnerable in the face of traumas such as wars and natural disasters, even choosing to end their own lives [1,2]. Several meta-analyses have examined suicide-related outcomes during infectious disease epidemics. The associations between epidemics and increased suicide risk are poorly supported [2,3,4], though a few review studies have reported higher suicide rates among older adults during SARS [4,5], more suicidal thoughts during an epidemic [2], and increased suicide attempts during SARS and Ebola [5]. It is unknown whether the COVID-19 pandemic and its consequences contribute to the rise of suicide risk. This study synthesized robust evidence to examine the potential changes in suicidal ideation, suicide attempt, and suicide before and during the COVID-19 pandemic.

The suicide risk was expected to be alarmingly severe over the short or long run following the outbreak of the COVID-19 pandemic, due to widespread and prolonged economic, social, health, and psychological vulnerability [6,7]. A few primary studies found an overall increase in the prevalence of suicidal ideation and attempts, and in the rate of death by suicide during the pandemic period compared with the pre-pandemic period [8,9,10], while other studies found a decreased trend [11,12,13], and some reported an overall stable trend [14,15,16]. Suicide studies during the pandemic tend to be methodologically poor [17], and high-quality evidence from an interrupted time-series study covering 33 countries showed no significant increases in suicide death in most countries/regions [18]. After synthesizing three studies, Prati et al. [19] found that the effect of the COVID-19 lockdown on suicide risk among the general population was not significant. Another meta-analysis [20] also indicated that the prevalence of suicidal behaviors and suicidal ideation did not increase significantly among youth from the general population and emergency department. In contrast, a living systematic review that included studies up to October 2020 found that the prevalence of suicidal ideation increased among COVID-19 patients, despite the fact that suicidal presentations in hospitals decreased [21]. Aligned with the upward trend found by a systematic review [22], the pooled prevalence of suicidal ideation and suicide attempt were estimated to be 10.81–12.1% and 4.86%, respectively [23,24], which were higher compared to the rates reported by pre-pandemic studies. Nevertheless, most of the existing reviews lack quantitative evaluation using intertemporal data [3,21,23], and the only meta-analysis comparing pre- and peri-pandemic prevalence included only young people [20]. As suicide outcomes may vary across populations while the pandemic progresses, it is necessary to keep track of suicidality and monitor the impact of the pandemic on suicide-related outcomes [17,25]. 

Suicide risk varies according to biological, clinical, psychological, social, cultural, and environmental factors [26,27]. For example, age and sex has shown mixed effects, so there is no consensus regarding whether young people or females are at a higher risk for suicide than others [28]. Financial loss and unemployment status drives suicide-related consequences [29]. The reporting styles (e.g., self-report or clinical interview) affects the accuracy of outcomes [28], and different timeframes for evaluation (a past period when occurrence of symptoms was measured, e.g., the past two weeks or lifetime) can also lead to different results [30]. Democracy of the government and COVID-19 pandemic-related variables were risk factors for developing suicidal ideation and suicidal behaviors during the COVID-19 pandemic in previous meta-analyses [23,31]. Therefore, investigating the factors that make one more vulnerable to suicide in the face of the pandemic is crucial for informing suicide management under the pandemic context. 

This study aimed to (a) evaluate the impact of the COVID-19 pandemic on suicide by examining whether the prevalence of suicidal ideation and suicide attempt and the rate of suicide death change before and after the pandemic, and (b) to examine the potential moderators for effect sizes via subgroup analysis and meta-regression. Specifically, suicidal ideation is defined as thoughts about ending one’s own life with deliberate consideration (passive ideation) or the planning of possible techniques (active ideation); suicide attempt refers to an attempt to end one’s life that may lead to death; death by suicide is the act of intentionally causing one’s own death [32]. To investigate what factors could affect the trend of suicide-related outcomes, this meta-analysis examined the effect of study-level variables (i.e., study design, reporting style, timeframe for measurement, time for data collection, and risk of bias score), participant characteristics (i.e., population type and sex distribution), economic index (i.e., changes in gross domestic product and unemployment rate before and during the pandemic), and government-level COVID-19 index (i.e., resilience score, stringency index, containment and health index, and economic support index) on changes in the prevalence of suicidal ideation, suicide attempt, and death by suicide.

## 2. Methods

This study followed the Preferred Reporting Items for Systematic Reviews and Meta-Analysis (PRISMA) checklist (Appendix A) [33] and Meta-Analysis of observational studies in epidemiology guidelines [34]. The meta-analysis protocol was registered on PROSPERO (CRD42022326575; https://www.crd.york.ac.uk/prospero/display_record.php?ID=CRD42022326575; accessed on 8 December 2022).

### 2.1. Search Strategy

Two team members systematically searched for relevant literature in 13 electronic databases: PubMed, Web of Science, EMBASE, CINAHL, Academic Search Premier, PsycINFO, PsycARTICLES, Psychology and Behavioral Sciences Collection, WHO COVID database, China National Knowledge Infrastructure, Wanfang, and CQVIP. Preprint articles published on Medrxiv servers were also searched. The search covered literature published up to December 2022. The search terms were built following the CoCoPop mnemonic [35]. As we had no restrictions on population, there were no specific search terms for Pop: Condition (suicid* OR “suicidal ideation” OR “suicidal thoughts” OR “suicidal plan” OR “suicide attempt” OR “completed suicide” OR “death by suicide”) AND Context (“COVID*” OR coronavirus OR “2019-ncov” OR “SARS-CoV-2” OR “cov-19” OR “2019 pandemic”). Corresponding Chinese search terms were used in Chinese databases. In addition, the references of identified studies and relevant review articles were screened to expedite the identification of eligible research. An example of search can be found in Appendix A.

### 2.2. Inclusion and Exclusion Criteria

Inclusion criteria were that studies (a) reported the prevalence of at least one form of suicide-related outcome (i.e., suicidal ideation/thoughts, suicide attempt, and completed suicide), or sufficient information to compute these variables; (b) used a repeated cross-sectional (i.e., pseudo-longitudinal; multiple assessments on different samples), longitudinal (multiple assessments on the same sample), or retrospective design providing at least one set of data for pre- and peri-pandemic periods (as defined by the study); (c) included measurement of suicidal ideation/thoughts and attempts of participants and/or analyzed country-level or regional data for suicide death among the general population; (d) were peer-reviewed journal articles or preprints with full text available; and (e) were written in English or Chinese.

Exclusion criteria were that studies (a) were review articles, case reports, commentary, books, conference papers, or other documents that did not present empirical findings with detailed method illustrations; (b) had no sufficient data to calculate the effect sizes; and (c) were duplicate sources.

### 2.3. Selection Procedure and Data Extraction

All the searched articles retrieved from the databases were imported to EndNote 20 for reference management. After removing the duplicates, the remaining articles written in English were exported to ASReview version 0.18. (https://asreview.nl/; accessed on 8 December 2022), an open-source machine learning program, for efficient title and abstract screening [36], and Chinese articles were screened manually. We selected the default combination of naive Bayes, maximization, and TF–IDF (term frequency–inverse document frequency) as the active learning model, which can produce consistently good results across many datasets [36]. To train the active learning model, the authors pre-selected relevant and irrelevant articles from the imported literature set. The ASReview presented the article titles and abstracts in order of relevance, and the authors continued to judge the relevance of articles successively until 50 consecutive irrelevant articles were marked in a row [37]. Finally, two team members applied the inclusion and exclusion criteria to independently review the full texts of the remaining studies to identify the eligible ones. Any disagreements between the two reviewers were resolved by discussion with the PI.

The following information was extracted: (a) identification of the study (i.e., title, first author’s name, publication year, country/region); (b) methodological characteristics (i.e., study design, sample size, definitions of pre- and peri-pandemic period, assessment approach for suicidal ideation and attempt/sources of death data,); (c) sample characteristics (i.e., population type, age and female proportion); and (d) outcome (i.e., prevalence/number of participants reporting suicidal ideation and/or suicide attempt for each period, or prevalence ratio comparing pre- and peri-pandemic assessment, rate/number of suicide death for both periods, or rate ratio comparing pre- and peri-pandemic data). Coding information varied slightly for studies reporting suicidal ideation, attempt, and death by suicide. In case of insufficient information on the published article, we contacted the authors via email. To consider financial factors, we retrieved data of gross domestic product/gross state product and unemployment rates [38,39,40,41] for both the pre- and peri-pandemic periods as defined by each included study, and calculated the ratio (peri/pre) to see how changes in financial factors would link to the variation in suicide rates. In addition, to monitor the impact of government reaction on suicide rate, we derived a resilience score (defined as an average score of reopening progress, COVID status, and quality of life in a country during the pandemic) from Bloomberg’s Covid Resilience Ranking [42] for the latest data (29 June 2022), and COVID-19 government response indexes (i.e., stringency index, containment and health index, and economic support index) from the COVID-19 Government Response Tracker [43] for the latest data upon our analysis (9 December 2022). Two team members coded the included studies independently, and discrepancies were resolved through discussion with the PI.

### 2.4. Risk of Bias Assessment

The Joanna Briggs Institute (JBI) Critical Appraisal Instrument for prevalence studies was used to assess the risk of bias [35]. This instrument consists of nine items examining bias from a few factors (e.g., sample frame, sampling method, sample size, sample description, data analysis, measurement scale, and response rate). Each item can yield a score of 1 (the absence of bias) or 0 (the presence of bias). The total score ranged from 0 to 9, with a higher score indicating a lower risk of bias. Two members rated each study independently, and all disagreements between the two raters were resolved via discussion with the PI.

### 2.5. Statistical Analysis

Prevalence ratio (PR) or rate ratio (RR) represented the measure of effect size to examine the changes between the pre- and peri-pandemic periods in suicidal ideation, suicide attempt, and death by suicide. Specifically, prevalence ratios for suicidal ideation and attempt were calculated using event prevalence/count and sample size before and during the pandemic. For suicide death data, single estimate of the rate ratio with 95% CI for every included sample can be (a) directly extracted from publications or (b) calculated using rates of suicide death per 100,000 people in pre- and peri-pandemic periods [44]. To accommodate different types of input data, Comprehensive Meta-Analysis version 2.0 was used to conduct meta-analysis. 

A random-effects model was used to pool PR or RR reported by each sample. We employed an index of Cochran’s *Q*, *Tau*^2^, and *I*^2^ statistics to test heterogeneity, with a *p* value of <0.05 for *Q*, *Tau*^2^, and *I*^2^ > 50% indicating significant between-study heterogeneity [45]. Publication bias was determined through visual inspection of the asymmetry of the funnel plot and Egger’s regression test [46]. Sensitivity analysis was conducted by omitting studies one by one (leave-one-out method) and excluding the studies with quality score ranked below 25% of the total before recalculating the pooled estimate, to determine the robustness of results [47].

Among our included samples, there were unneglectable heterogeneities regarding clinical and methodological characteristics. Some of the samples consisted of participants recruited from non-clinical settings (e.g., college or general population), and these were mainly prospective studies using self-report methods to measure suicidal outcomes, while the other samples were from clinical settings (e.g., emergency or psychiatric departments) whose suicide-related data were retrospectively extracted from medical diagnosis. Based on previous studies, the prevalence of suicidal ideation and suicide attempt can be different between individuals from non-clinical settings and clinical settings during both pre- and peri-pandemic periods [20,21,48,49]. Considering these differences in physical or mental health conditions, patterns of changes in suicide-related outcomes, and several methodological characteristics, the prevalence ratio for suicidal ideation and suicide attempts was analyzed by non-clinical and clinical samples. In general, the non-clinical sample refers to participants recruited from community and non-medical settings (e.g., college students and the general population), and the clinical sample refers to participants recruited from medical settings (e.g., patients from the emergency and psychiatric departments).

To determine the source of heterogeneity, we conducted subgroup analysis and meta-regression. Specifically, we used a mixed effect model in subgroup analysis, where a random-effects model was used to pool samples within each subgroup, and a fixed effect model was used to pool the subgroup to yield the overall estimates. Meta-regression was run under the random-effects model to examine the effects of continuous variables on ratio. For studies reporting the pre- and peri-pandemic prevalence of suicidal ideation and/or attempt, population group (adolescent, younger group, general population/adult, or special group), study design (repeated cross-sectional, longitudinal, or retrospective), method (self-report or diagnosis) and timeframe (≤2 weeks or >2 weeks) for measuring suicidal outcomes, data collection (March–August 2020, September 2020–January 2021, or February 2021+), female percentage and risk of bias score were considered as potential moderators for effect sizes. Specifically, time for data collection was divided into 6-month intervals beginning from the onset of the pandemic to capture the changes in the global pandemic situation according to data from the World Health Organization [50]. In addition, we examined the impact of changes in GDP (Peri/Pre), unemployment rate (Peri/Pre), and several government-level COVID-19 indexes (i.e., resilience score, stringency index, containment and health index, economic support index) on the ratio of death by suicide.

## 3. Results

### 3.1. Study Characteristics

A total of 8642 studies were screened, with 72 studies (131 samples) meeting the inclusion criteria. Among them, 16 [10,11,51,52,53,54,55,56,57,58,59,60,61,62,63,64], 11 [65,66,67,68,69,70,71,72,73,74,75], and 19 [8,13,14,76,77,78,79,80,81,82,83,84,85,86,87,88,89,90,91] only addressed changes in suicidal ideation, suicide attempt, and death by suicide, respectively; 25 [9,12,15,16,92,93,94,95,96,97,98,99,100,101,102,103,104,105,106,107,108,109,110,111,112] reported both changes in suicidal ideation and attempt, and one study [113] reported both changes in death and attempt. The flow diagram of the study is depicted in Figure 1. The specific research information for ideation and attempt samples and death samples is shown in Table 1 and Table 2, respectively. Considering the unneglectable differences between non-clinical and clinical conditions, results for suicidal ideation and suicide attempt were reported separately based on the study settings. Specifically, the non-clinical sample was composed of adolescents, high school students, college students, the general population, hotline callers and military veterans, as stated by each included study; the clinical sample included pediatric and adult patients from emergency, psychiatric emergency, and inpatient departments, and those with certain medical conditions (i.e., obesity and eating disorder), as specified by each included study.

### 3.2. Meta-Analysis of Suicidal Ideation for Non-Clinical and Clinical Samples

#### 3.2.1. Overview

There were 41 studies with 51 samples (28 non-clinical and 23 clinical samples, respectively) included for the analysis of suicidal ideation. Specifically, several studies contributed more than one sample due to multiple peri-pandemic data collection [9,54,59,62,64,106,108] or subsets of participants [103,111]. With an average risk-of-bias score of 7.2 (range = 5–9), the majority of included studies had adequate size and information for the sample, addressed response rate properly, and employed appropriate statistical methods. The most common limitations were deficiencies in the sample representativeness and recruiting methods, and unclear measurements (Appendix A). Both non-clinical and clinical settings showed an increased prevalence of suicidal ideation during the pandemic compared with pre-pandemic periods, and significant heterogeneity was found within each setting (Table 3).

#### 3.2.2. Non-Clinical Samples

The point estimates of suicidal ideation reported by 28 non-clinical samples ranged from 0.332 to 4.794, and the pooled prevalence ratio under the random effect model was 1.142 (95% CI: 1.018–1.282; *p* = 0.024), indicating a higher prevalence of suicidal ideation during the pandemic compared with the pre-pandemic period (Figure 2). The heterogeneity test results were significant (*I*^2^ = 97.734%, *tau*^2^ = 0.081, *P_Q_* < 0.05), indicating that there was a large difference in effect sizes between samples. Sensitivity analysis using leave-one-out method showed that most of the included samples did not affect the outcome substantially. However, when Ettman et al. [10] and Kasal et al. [9] were excluded separately, the respective pooled ratio (1.007–1.112) indicated a slight but non-significant increase in the peri-pandemic prevalence of suicidal ideation compared with pre-pandemic periods (Appendix A). No publication bias was observed among the non-clinical samples according to the funnel plot (Appendix A) and the non-significant results from Egger’s tests (*intercept* = 2.61, *t* = 1.48, *p* = 0.151).

The results of subgroup analysis are shown in Table 4; they suggest that the prevalence ratio varied by population type and study design. The highest PR was found among the general population (PR = 2.014, 95% CI: 1.604–2.529; *p* = 0.000), which indicated that suicidal ideation was twice as prevalent during the COVID-19 pandemic compared with before, while the prevalence of suicidal ideation in adolescents, younger (mostly college students) and special populations remained basically unchanged. After excluding the only one study [58] using retrospective design, the prevalence ratio was 1.318 (95% CI: 1.132–1.535; *p* = 0.000) for repeated cross-sectional studies, suggesting the significantly increased prevalence of suicidal ideation during the pandemic relative to pre-pandemic times; meanwhile, longitudinal studies showed a non-significant decrease (PR = 0.842; 95% CI: 0.666–1.063; *p* = 0.148). Meta-regression showed that neither female percentage nor quality score for non-clinical samples was associated with PR (Appendix A).

#### 3.2.3. Clinical Samples

Of the 51 samples reporting suicidal ideation, 23 were conducted in a clinical setting. With a pooled estimate of 1.134 (95% CI: 1.048–1.227; *p* = 0.002), the prevalence ratio for each study ranged from 0.177 to 2.262 (Figure 3). As indicated by the results, there was significant heterogeneity among the samples (*I*^2^ = 71.029%, *tau*^2^ = 0.018, *P_Q_* < 0.05). By excluding samples one by one or samples with lower quality [62,96,107], sensitivity analysis showed that none of the samples affected the outcome substantially (Appendix A). Thus, despite variations across studies, the pooled estimate was robust enough to show an increasing trend of suicidal ideation among clinical patients during the pandemic. Through visual inspection of the funnel plot (Appendix A) and the non-significant results in the Egger’s tests (*intercept* = 0.096, *t* = 0.15, *p* = 0.882), the results showed that there was no publication bias among the clinical samples.

As all the clinical samples employed a retrospective design by extracting medical records, the subgroup analysis considered only population (adolescent vs. adult patients), method, and timeframe for measurement tool. According to Table 4, none of the above variables was a significant moderator for PR in the clinical samples. Nevertheless, meta-regression (Appendix A) showed that study quality was positively associated with the ratio, suggesting that higher-quality studies tended to report a larger increase in suicidal ideation (*B* = 0.08, *p* < 0.05).

### 3.3. Meta-Analysis of Suicide Attempt for Non-Clinical and Clinical Samples

#### 3.3.1. Overview

There were 37 studies with 55 samples (30 non-clinical and 25 clinical samples, respectively) included for the analysis of suicidal ideation. Specifically, several studies contributed more than one sample due to multiple peri-pandemic data collection [9,65,70,75,106,108] or subsets of participants [75,103,111]. With an average risk of bias score of 7.5 (range = 4–9), the majority of included studies had adequate size and information for their sample, and they employed appropriate sampling and statistical methods. The most common limitations were deficiencies in sample representativeness and unclear criteria for judging suicide attempt (Appendix A). Both non-clinical and clinical settings showed increased suicide attempts during the pandemic, compared with pre-pandemic periods, and the increase was higher among clinical participants. Significant heterogeneity was found in each setting (Table 5).

#### 3.3.2. Non-Clinical Samples

The prevalence ratio of suicide attempt reported by 30 non-clinical samples ranged from 0.333 to 6.261, and the pooled prevalence ratio under the random effect model was 1.14 (95% CI: 1.053–1.233; *p =* 0.001), indicating that suicide attempts were more prevalent during the COVID-19 pandemic than during pre-pandemic periods (Figure 4). Though effect sizes were substantially heterogenous among the samples (*I*^2^ = 99.996%, *tau*^2^ = 0.036, *P_Q_* < 0.05), a sensitivity analysis using the leave-one-out method or by excluding lower quality studies [66] showed that the increased trend of suicide attempts was robust (Appendix A). Through visual inspection of the funnel plot (Appendix A) and the non-significant results of the Egger’s tests (*intercept* = 6.52, *t* = 0.17, *p* = 0.865), no publication bias among the non-clinical samples was observed.

The results of the subgroup analysis are shown in Table 6. Excluding only one prisoner sample, the general population (PR = 1.218, 95% CI: 1.089–1.362; *p* = 0.001) showed the largest increase in suicide attempts compared with adolescent and younger samples, despite the fact that the differences were insignificant. Meta-regression showed that neither the female percentage nor the quality score for the non-clinical samples were associated with PR (Appendix A).

#### 3.3.3. Clinical Samples

Of the 55 samples reporting suicide attempt, 25 were conducted in a clinical setting. With a pooled estimate of 1.32 (95% CI: 1.17–1.489; *p* = 0.000), the prevalence ratio for each study ranged from 0.71 to 2.379 (Figure 5), and the effect size was heterogenous (*I*^2^ = 70.021%, *tau*^2^ = 0.052, *P_Q_* < 0.05). The increased trend for suicide attempt during the pandemic was robust, as pooled estimates did not change substantially based on the results (Appendix A) of the leave-one-out sensitivity analysis and the analysis excluding lower quality studies [96,107]. Visual inspection of the funnel plot (Appendix A) and the non-significant results in the Egger’s tests (*intercept* = −0.63, *t* = 0.86, *p* = 0.397) indicated an absence of asymmetry in the funnel plot. These results showed no publication bias among the clinical studies reporting suicide attempt.

All clinical samples (*k* = 25) employed a retrospective design by extracting medical records; subgroup analysis considered only population (adolescent or adult patients), method and timeframe for measurement tool, and time for data collection. According to subgroup analysis (Table 6) and meta-regression (Appendix A), no significant moderators were found for PR of suicide attempts in the clinical samples.

### 3.4. Meta-Analysis for Death by Suicide

A total of 25 samples were reported by 20 studies, as some studies included gender-specific [81] or multiple peri-pandemic [84,89,114] data. The risk-of-bias score for the included samples ranged from 5 to 9 (average = 7.8), and 75% of the samples scored above 7. All samples were nationally or regionally representative, while some of them did not provide detailed demographics, criteria for judging, or source of suicide death (Appendix A). 

As shown in Figure 6, the point estimate ranged from 0.429 to 1.207, and the pooled rate ratio for suicide death was 0.923 (95% CI: 0.84–1.01; *p* > 0.05) under the random-effect model. The heterogeneity test results were significant (*I*^2^ = 99.99%, *tau*^2^ = 0.055, *P_Q_* < 0.05), indicating a substantial difference in effect sizes across samples. By excluding samples one by one or samples with lower quality scores [14,77,78,83,86], sensitivity analysis showed robustness in the pooled ratio for death by suicide, as none of the samples affected the pooled estimate substantially (Appendix A). In addition, visual inspection of the funnel plot (Appendix A) and the non-significant results in the Egger’s tests (*intercept* = 11.99, *t* = 0.21, *p* = 0.837) indicated an absence of asymmetry in the funnel plot. These results showed that there was no publication bias when deriving the pooled estimate.

Meta-regression was conducted to test whether counties’ resilience and government-level economic and societal indexes during the pandemic would contribute to the between-study heterogeneity on death by suicide. However, the moderating effects for these indexes were not significant (Appendix A). The subgroup analysis showed that the trends for suicide death were significantly different between national and regional samples (*p* = 0.01). Despite the country-level rate of death by suicide remaining stable (RR = 0.99, 95% CI: 0.91–1.1; *p* > 0.05), data from regional samples reported a decreased trend (RR = 0.82, 95% CI: 0.73–0.92; *p* = 0.001).

## 4. Discussion

To our knowledge, this work was the first meta-analysis that assessed the changes in the prevalence of suicidal ideation, suicide attempt, and rate of death by suicide before and during the COVID-19 pandemic across populations, using intertemporal data from repeated cross-sectional retrospective, longitudinal, and retrospective studies. We included 45 studies with 67 samples, and most of the included studies had a low risk of coverage bias, sample size estimation, and statistical analysis. Compared with the pre-pandemic period, the prevalence of suicidal ideation and suicide attempt increased significantly during the COVID-19 pandemic among both non-clinical and clinical samples, while the rate of death by suicide remained mostly unchanged in the synthesis of the existing evidence. 

Our findings showed an upward trend of suicidal ideation and suicide attempt during the COVID-19 pandemic among both non-clinical and clinical samples. These results were consistent with a few studies during the pandemic, which warned about the increased risks of suicidal thoughts and behaviors relative to pre-pandemic periods among the general population and inpatients [22,23,24]. The outbreak of the COVID-19 pandemic had brought profound health, psychological, social, and economic consequences worldwide, which might have heightened various suicide risk factors [6,7]. As found by a meta-analysis looking at data spanning 50 years [115], hopelessness, mental health issues (e.g., depression and anxiety), socioeconomic status, and stressful life events were among the top predictors for suicide-related outcomes. These factors were also applied to the pandemic context [24]. During the pandemic, with lockdown measures implemented, individuals experienced overwhelming fears and worries about COVID-19 due to health issues, uncertainties about the future, stigmatization, and misinformation from media, which were associated with higher hopelessness under the pandemic context [116]. Additionally, the COVID-19 pandemic had been described as a tsunami, leading to mental disorders worldwide [117], and a wide range of studies had suggested the severe psychological impacts of increased distress, depression, anxiety, insomnia, and loneliness brought by the pandemic across populations [118,119,120,121]. These psychological symptoms might be long-lasting, increasing the risk for suicidality [7]. People also suffered from financial strain, unemployment, and economic uncertainty due to the global economic downturns, which constituted societal risk factors for developing suicidal ideation and attempts during the pandemic [122]. Recent studies have also reported the rise of other risk factors for suicide, such as weakened social support, poor health, increased interpersonal conflict, domestic violence, and alcohol consumption [6,123,124,125,126]. The wide-ranging adverse effects of the pandemic may have put individuals at a disadvantage and triggered the increased suicide-related outcomes. 

Notably, the adult general population showed a larger increase in both suicidal ideation and suicide attempt, and the increase in suicidal ideation was particularly noticeable. The PR for suicidal ideation was 2.014 (95% CI: 1.604–2.529; *p* = 0.000), indicating a doubled prevalence of suicidal ideations during the pandemic, with a slight increase in suicide attempt (PR = 1.218; 95% CI: 1.089–1.362; *p* = 0.001). The adult general population in our study was aged above 25 and mostly in their 30s–50s. Middle-aged adults are usually the pillar of a family, with heavier financial and caregiving responsibilities. Thus, economic adversities and lockdown measures can threaten this population, making them more vulnerable to suicide [127]. In addition, the larger increase in ideation than attempt echoes the pyramid theory of suicidal trajectories [128]. According to the theory, suicide-related outcomes develop in an ascending flow from suicidal ideation at the bottom of the pyramid, moving up to plan and attempt, and ultimately reaching the peak of the pyramid, completed suicide. Individuals can stop at any stage once they have started the “suicidal career”, but most people only have suicidal thoughts, with few actually taking action [129]. The gap between these stages can be understood by the interpersonal theory of suicide, which suggests that despite a strong suicidal desire, the step toward attempt requires one’s ability (e.g., fearlessness and pain insensitivity) to act on the thoughts [130]. Thus, not everyone who develops suicidal ideation engages in suicidal behaviors, which can account for the differences in incremental movement from suicidal ideation to attempt during the pandemic among the non-clinical adult samples. 

Interestingly, our clinical samples showed more increases in the prevalence of suicide attempt than suicidal ideation. These findings do not contradict the pyramid theory. Previous studies found that suicidal thoughts were often under-documented in clinical settings [131], and those who attempted suicide were more prone to present in emergency and psychiatric departments compared to those only with suicidal ideation. In addition, referral to health services was further hindered by the lockdown measures during the pandemic, resulting in a larger proportion of community individuals with only suicidal thoughts being underdiagnosed [132,133]. These reasons can also explain our results that suicidal ideation among non-clinical adults increased more than the clinical samples, which is congruent with a previous finding [20]. 

The results for suicidal ideation and attempt were mostly robust, based on the sensitivity analysis. The only exception was found by excluding several non-clinical samples one by one [9,10], the pooled estimates which changed substantially, and the fact that the increases in suicidal ideation became smaller and nonsignificant, indicating the peculiarly large increases in these studies. Both studies employed a repeated cross-sectional design; therefore, those who responded to the survey in two periods may have differed in characteristics. Though both studies collected data from nationwide adult samples during the pre- and peri-pandemic periods, participants recruited during the pandemic were experiencing more adverse conditions for suicidality, as mentioned above (e.g., mental disorders, economic disadvantage). This finding again suggests that suicide risk factors were magnified by the pandemic, and the adult population may have suffered more. In any case, our subgroup analysis among non-clinical samples showed that, compared with the longitudinal study, suicidal ideation reported by repeated cross-sectional studies increased more. Therefore, it is possible that stressors may have affected survey participation [10], as psychologically vulnerable individuals may pay more attention to mental health information during the pandemic due to attentional bias. 

Suicide death did not change significantly in terms of the pooled RR during the pandemic, compared with the pre-pandemic period. This trend agrees with previous findings from 33 countries and individual groups [18,20]. However, the trend was different at the regional and country level, with a significant downward trend shown by using regional data (RR = 0.82, 95% CI: 0.73–0.92; *p* = 0.001). The difference was also found in the state of Connecticut, showing a lower suicide rate compared with the national level [82]. The possible explanation might be the small sample size or the larger coverage of a single race included by regional data, which is not representative of the national profile. This implies the need to consider regional differences and the representativeness of the samples when interpreting the suicide rate. 

Our findings have significant implications for future suicide management. Although the overall rate of death by suicide did not increase, suicide concerns are still serious, as this study showed that suicidal ideation and suicide attempts have been more prevalent since the pandemic. Having suicidal desires and acting on the thoughts are the prior stages of final death by suicide; such suicidal processes can be unstable and vary in duration. For example, the average duration for females and males before displaying explicit suicidal acts was 52 and 31 months [129]. In other words, a “suicidal career” takes time to progress to the final stage, though this only applies to a small proportion of suicidal individuals [134]. Thus, the alarming increases in suicidal ideation and suicide attempts during the pandemic point to the need for prompt suicide screening and prevention—especially among the members of the general public who might be underdiagnosed—and specifically, timely interventions targeting suicidal individuals to halt their exacerbation. 

This study has a few limitations. First, some included studies did not provide sufficient information (e.g., gender distributions and timeframe for measurement) to be coded for the moderation analysis, making our subgroup analysis and meta-regression results less convincing. Second, there may have been an overlap in the subgroup of the adult general population and younger group (mostly college students), as the studies targeting the general population usually included people above 18, despite the larger proportion being middle-aged. Third, considerable heterogeneity still exists in all outcomes, even after considering potential moderators, and none of the investigated factors can account for the variability among effect sizes for death by suicide, which was similar to the previous findings [18]. Future studies are recommended to examine other potential sources of heterogeneity. Finally, the included samples only covered data up to November 2021, most of which were conducted in 2020. Such a delay may have compromised the validity of our findings, as suicidality and its risk factors are fluid in nature and vary within short periods, according to the fluid vulnerability theory of suicide [135]. As the pandemic evolves, the present suicide situation might be changing, so it is necessary to have ongoing monitoring and real-time surveillance. 

## 5. Conclusions

In conclusion, our study provides an overview of the changes in the prevalence of suicidal ideation and suicide attempts across populations and the national or regional rate of death by suicide since the outbreak of the COVID-19 pandemic. Although the overall rate of death by suicide remained basically unchanged during the pandemic, suicidal ideation and suicide attempt were more prevalent compared with the pre-pandemic period, especially among the adult general population and clinical patients. Considering the heightened suicide risk factors, such as mental health problems and economic vulnerability during the pandemic, large-scale suicide screening for the public and timely intervention programs for high-risk groups are highly needed. The continuously changing pandemic underscores the importance of ongoing monitoring and surveillance for suicidality.

## Figures and Tables

**Figure 1 ijerph-20-03346-f001:**
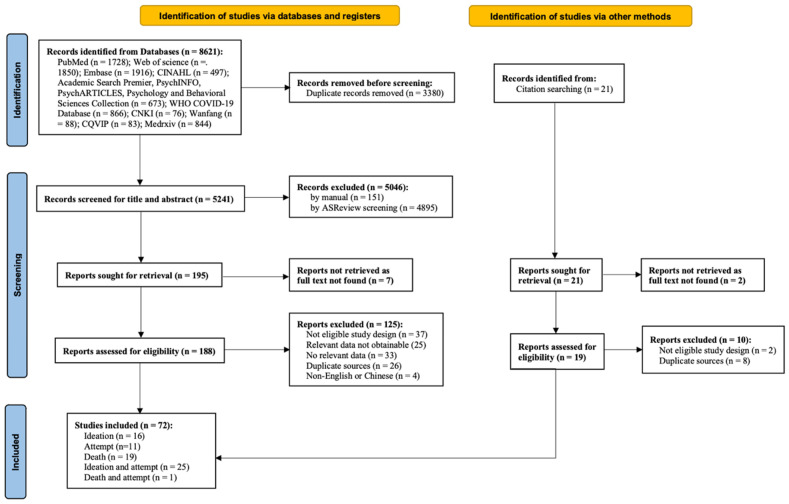
PRISMA flow chart.

**Figure 2 ijerph-20-03346-f002:**
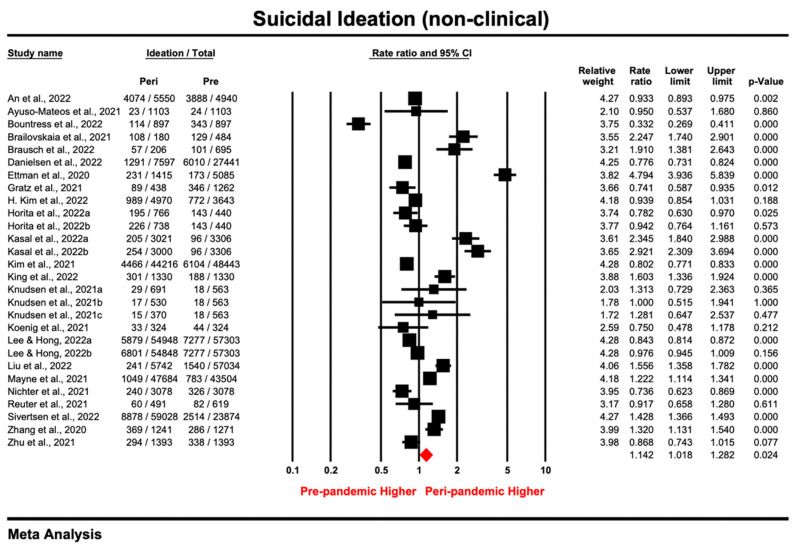
Forest plot for suicidal ideation of studies [9,10,11,12,15,51,52,53,54,55,56,57,58,61,64,93,97,98,99,100,102,108,112] conducting in the non-clinical setting (Prismatic colored as red refers to the pooled estimate).

**Figure 3 ijerph-20-03346-f003:**
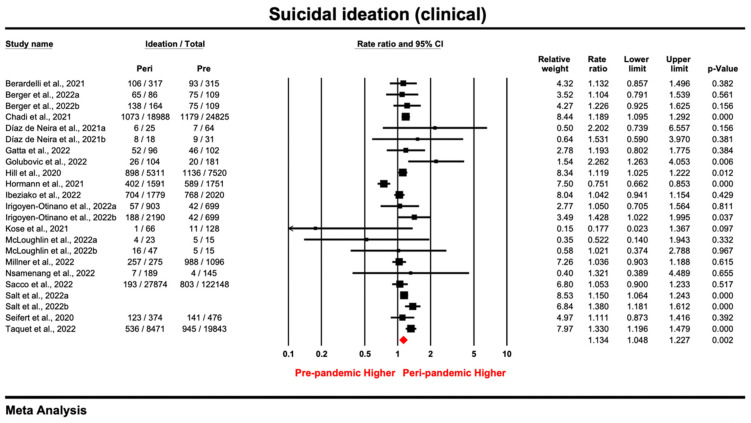
Forest plot for suicidal ideation of studies [16,59,60,62,63,92,94,95,96,101,103,104,105,106,107,109,110,111] conducted in the clinical setting (Prismatic colored as red refers to the pooled estimate).

**Figure 4 ijerph-20-03346-f004:**
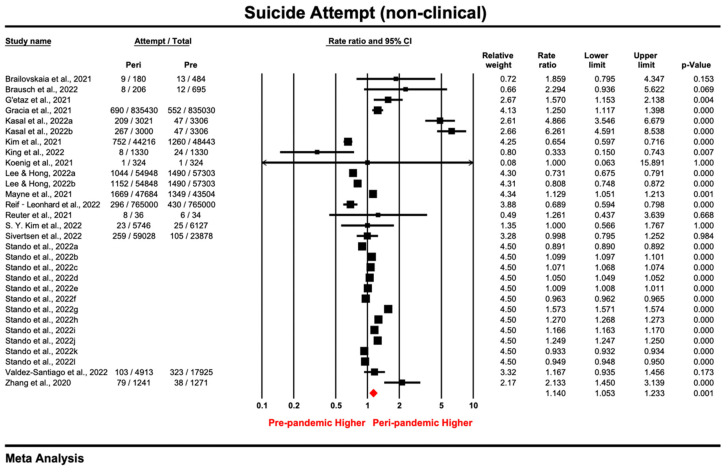
Forest plot for suicide attempt of studies [9,12,15,66,68,72,74,75,93,97,98,99,100,102,108,112,113] conducted in the non-clinical setting (Prismatic colored as red refers to the pooled estimate).

**Figure 5 ijerph-20-03346-f005:**
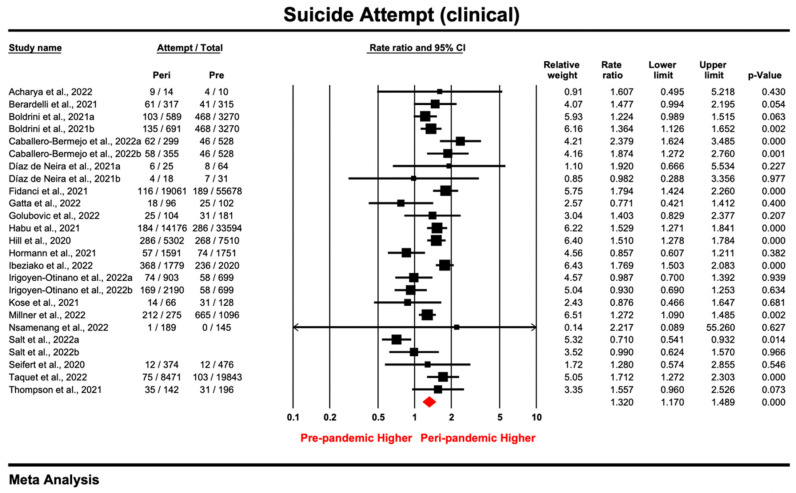
Forest plot for suicide attempt of studies [16,65,67,69,70,71,73,92,94,95,96,101,103,104,105,106,107,109,110,111] conducted in the clinical setting (Prismatic colored as red refers to the pooled estimate).

**Figure 6 ijerph-20-03346-f006:**
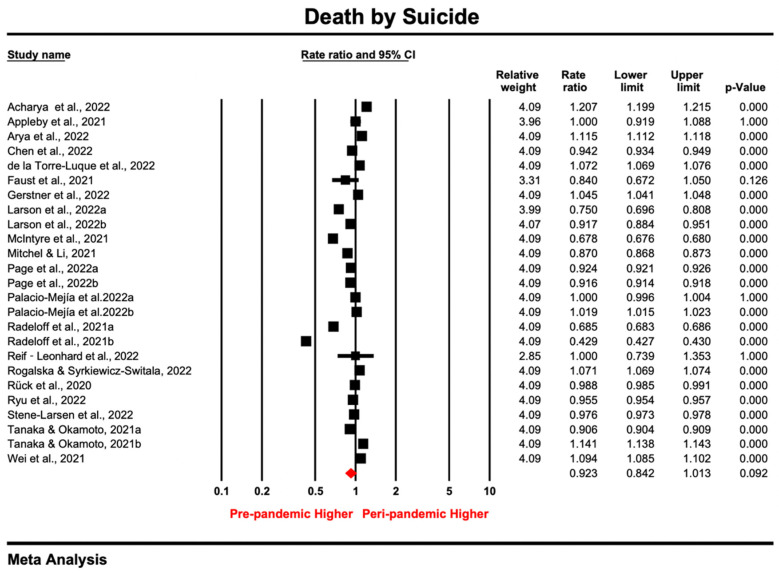
Forest plot for studies [8,13,14,76,77,78,79,80,81,82,83,84,86,87,88,89,90,91,113,114] reporting death by suicide (Prismatic colored as red refers to the pooled estimate).

**Table 1 ijerph-20-03346-t001:** Characteristics for studies reporting suicidal ideation and suicide attempt.

Study	Country	Study Design	Population Type	Sample Size	Participant Characteristic		Time Range		Outcome	Measurement
					Mean Age/Age Group	Female%	Pre-Pandemic Period	Peri-Pandemic Period		
Acharya et al., 2022 [69]	India	Retrospective	Indoor Patients With Cut Throat Injury	Pre (10); Peri (14)	25 (N = 17); 35 (N = 2); 45 (N = 4); 55 (N = 1)	4.16%	Sept. 1st, 2019–Feb. 28th, 2020	Mar. 1st–Aug. 31st for 2020	Attempt	Diagnosis (Identified by tentative cut mark)
An et al., 2022 [58]	China	Retrospective	Hotline Callers	Pre (4940); Peri (5550)	Pre: <30 (76%), >30 (24%); Peri: <30 (74.4%), >30 (25.6%)	Pre (52.7%); Peri (60.6%)	Jan. 21st–Jun. 30th for 2019	Jan. 21st–Jun. 30th for 2020	Ideation	Self-report (Assessed by asking the caller Have you repeatedly thought about taking your life or hurting yourself in the last two weeks? Or have you felt too tired and without meaning to continue to live in the last two weeks?
Ayuso-Mateos et al., 2021 [51]	Spain	Longitudinal	General Population	1103	54.82	64.8%	Jun. 17th 2019–Mar. 14th 2020	May 21st 2020–Jun. 30th 2020	Ideation	Self-report (An item of Composite International Diagnostic Interview: Whether the participant had had suicidal thoughts in the previous 12 months/30 days)
Berardelli et al., 2021 [92]	Italy	Retrospective	Psychiatric Patients	Pre (315); Peri (317)	42.25	49.2%	May 2019–Mar. 9th 2020	Mar. 10th–Dec. 2020	Ideation Attempt	Diagnosis (definition: thoughts about wishing to be dead or active thoughts of wanting to end one’s life) Diagnosis (Definition: a non-fatal, self-directed, potentially injurious behavior with an implicit or explicit intent to die; the behavior may or may not result in injury, and the intensity may vary, but the decision to act out the lethal intent must be present)
Berger et al., 2022 [59]	Switzerland	Retrospective	Emergency Psychiatric Adolescent Patients	Pre (109); Peri 1 (86); Peri 2 (164)	Pre (14.89); Peri 1 (14.81); Peri 2 (14.41)	Pre (56.6%); Peri 1 (57%); Peri 2 (64.2%)	Mar. 1st–Apr. 30th for 2019	Mar. 1st–Apr. 30th for 2020; Mar. 1st–Apr. 30th for 2021	Ideation	Diagnosis (Medical record)
Boldrini et al., 2021 [65]	Italy	Retrospective	Psychiatric Patients	Pre (3270); Peri 1 (589); Peri 2 (691)	45.4	52.70%	March 1st–Jun. 30th for 2018–2019	Mar. 1st–Apr. 30th for 2020; May 1st–Jun. 30th for 2020	Attempt	Diagnosis (Medical record)
Bountress et al., 2022 [52]	United State	Longitudinal	College Students	897	18.49 (Baseline)	78.6%	Spring of 2019	May 7th 2020–Jul. 17th 2020	Ideation	Self-report (a Symptom Checklist-90 Revised: participants answered whether they had thought about killing themselves by yes or no).
Brailovskaia et al., 2021 [93]	Germany	Repeated Cross-sectional	College Students	Pre: 2016 (105); 2017 (117); 2018 (108); 2019 (154); Peri: 2020 (180)	Pre: 2016 (22.51); 2017 (22.53); 2018 (20.59); 2019 (21.98); Peri: 2020 (21.33)	Pre: 2016 (81.9%); 2017 (78.6%); 2018 (73.1%); 2019 (84.4%); Peri: 2020 (73.3%)	Oct.–Dec. for 2016–2019	Oct.–Dec. for 2020	Ideation Attempt	Self-report (a Suicidal Behaviors Questionnaire-Revised: How often have you thought about killing yourself in the past year?) Self-report (A Suicidal Behaviors Questionnaire-Revised: Have you ever attempted suicide, and really hoped to die?)
Brausch et al., 2022 [102]	United State	Repeated Cross-sectional	High School Students	Pre (695); Peri (206)	Pre (15.5); Peri (15.6)	Pre (54.8%); Peri (57.8%)	2018–2019	2020–2021	Ideation Attempt	Self-report (self-injurious thoughts and behaviors interview—short form; past year)
Caballero-Bermejo et al., 2022 [70]	Spain	Retrospective	Emergency Patients	Pre (528); Peri 1 (299); Peri 2 (355)	Pre (31.4); Peri 1(41.3); Peri 2 (38.3)	Pre (41.5%); Peri 2 (33.1%); Peri 2 (41.4%)	Jun.–Jul. for 2019	Jun.–Jul. for 2020; Jun.–Jul. for 2021	Attempt	Diagnosis (medical record)
Chadi et al., 2021 [60]	Canada	Retrospective	Emergency Adolescent Patients	Pre (24,824.5); Peri (18,988)	--	--	2018–2019	2020	Ideation	Diagnosis (International Classification of Diseases, 10th edition)
Danielsen et al., 2022 [11]	Denmark	Longitudinal	Young People	Pre (27,441); Peri (7597)	20.7 (Wave 8)	Pre (56%); Peri (67%)	Jan. 1st 2018–Mar. 11th for 2020	Mar. 12th 2020–Mar. 1st 2021	Ideation	Self-report (Have you thought about taking your own life, even though you would not do it, within the last year?)
Díaz de Neira et al., 2021 [103]	Italy	Retrospective	Emergency Adolescent Patients; Psychiatric Adolescent Inpatients	Emergency sample: Pre (64); Peri (25); Psychiatric sample: Pre (31); Peri (18)	Emergency sample: Pre (14.2); Peri (15.36); Psychiatric Sample: Pre (15.55); Peri (15.17)	Emergency sample: Pre (62.5%); Peri (72%); Psychiatric Sample: Pre (64.5%); Peri (72.2%)	2019	2020	Ideation Attempt	Diagnosis (medical record)
Ettman et al., 2020 [10]	United State	Repeated Cross-sectional	General Population	Pre (5085); Peri (1415)	Pre (46.57); Peri (45.62)	Pre (51.3%); Peri (50%)	2017–2018	Mar. 31st 2020–Apr. 13th 2020	Ideation	Self-report (Item 9 of the Patient Heath Questionarie-9: Thoughts that you would be better off dead or of hurting yourself in some way over the past two weeks?)
Fidancı et al., 2021 [71]	Turkey	Retrospective	Pediatric Emergency Adolescents	Pre (55,678); Peri (19,061)	Pre (8.11); Peri (8.58)	Pre (47.6%); Peri (48.9%)	Apr.–Oct. for 2019	Apr.–Oct. for 2020	Attempt	Diagnosis (medical record)
Gatta et al., 2022 [94]	Italy	Repeated Cross-sectional	Psychiatric Adolescent Patients	Pre (102); Peri (96)	Pre (13.2); Peri (13.8)	Pre (63.7%); Peri (65.6%)	Feb. 2019–Feb. 2020	Mar. 2020–Mar. 2021	Ideation Attempt	Diagnosis (medical record)
G’etaz et al., 2021 [66]	Switzerland	Retrospective	Prisoners	--	--	--	2016–2019	2020	Attempt	Diagnosis (medical record)
Golubovic et al., 2022 [104]	Serbia	Retrospective	Psychiatric Adult patients	Pre (181); Peri (104)	40.58	41.55%	May–Aug. for 2018–2019	May–Aug. for 2020	Ideation Attempt	Diagnosis (definition: thoughts about wishing to be dead or active thoughts of wanting to end one’s life) Diagnosis (definition: nonfatal, self-directed, potentially injurious behavior with an implicit or explicit intent to die).
Gracia et al., 2021 [72]	Spain	Retrospective	Adolescent	Pre (835,030); Peri (835,430)	--	--	Mar. 2019–Mar. 2020)	Mar. 2020–Mar. 2021	Attempt	Diagnosis (Six-item suicidality module of the Mini International Neuropsychiatric Interview)
Gratz et al., 2021 [53]	United State	Repeated Cross-Sectional	College Students	Pre1 (539); Pre2 (723); Peri (438)	Pre1 (19.39); Pre2 (19.26); Peri (19.53)	Pre1 (69.2%); Pre2 (64.9%); Peri (61.9%)	Fall 2014; Fall 2013	Fall 2020	Ideation	Self-report (pre: How often have you thought about killing yourself in the past year?; peri: Have you had thoughts of killing yourself in the past year?).
H. Kim et al., 2022 [55]	United State	Repeated Cross-Sectional Study	College Students	Pre (3643); Peri (4970)	Pre (18.85); Peri (19.5)	Pre (73.1%); Peri (69.7%)	Oct. 7th–Dec. 1st for 2019	Mar. 2nd–May 9th for 2020	Ideation	Self-report (Item 9 of the Patient Heath Questionarie-9: Thoughts that you would be better off dead or of hurting yourself in some way over the past two weeks?)
Habu et al., 2021 [73]	Japan	Retrospective	Emergency Call Patients	Pre (33,594); Peri (14,176)	Pre (60.1); Peri (62.8)	Pre (48.7%); Peri (49.2%)	Mar.–Aug. for 2018–2019	Mar.–Aug. for 2020	Attempt	Diagnosis (medical record)
Hill et al., 2020 [95]	United State	Retrospective	Pediatric Patients	Ideation: Pre (7520); Peri (5311) Attempt: Pre (7510); Peri (5302)	14.52	59%	Jan.–Jul. for 2019	Jan.–Jul. for 2020	Ideation Attempt	Self-report (Columbia-Suicide Severity Rating Scale)
Horita et al., 2022 [54]	Japan	Repeated Cross-Sectional	College Students	Pre (440); Peri 1 (766); Peri 2 (738)	--	Pre (51.4%); Peri 1 (45.3%); Peri 2 (49.5%)	Apr. 15th–May 31st for 2019	Apr. 20th–May 31st for 2020; Apr. 10th–May 31st for 2021	Ideation	Self-report (an item of the Counseling Center Assessment of Psychological Symptoms: I have had thoughts of ending my life in the past 2 weeks).
Hörmann et al., 2021 [96]	Switzerland	Retrospective	Psychiatric Patients	Pre (1751); Peri (1591)	Pre: <30 (22.8%), 30–59 (61.55), 60+ (15.7%); Peri: <30 (24.4%), 30–59 (59.8%), 60+ (15.8%)	Pre (46.2%); Peri (46.7%)	Mar. 13th–May 11th for 2019	Mar. 13th–May 11th for 2020	Ideation Attempt	Diagnosis (Arbeitsgemeinschaft für Methodik und Dokumentation in der Psychiatrie System)
Ibeziako et al., 2022 [105]	United State	Retrospective	Pediatric Psychiatric Patients	Pre (2020); Peri (1779)	Pre (8.11); Peri (8.58)	Pre (55.9%); Peri (65.8%)	Mar. 2019–Feb. 2020	Mar. 2020–Feb. 2021	Ideation Attempt	Diagnosis (International Classification of Diseases, 10th edition)
Irigoyen-Otiñano et al., 2022 [106]	Spain	Retrospective	Psychiatric Emergency Patients	Pre (699); Peri 1 (903); Peri 2 (2190)	Pre (43.6); Peri 1 (39.6); Peri 2 (37.1)	Pre (60%); Peri 1 (59.2%); Peri 2 (60.5%)	Jan. 13th–Mar. 14th for 2020	Mar. 15th–Jun. 20th for 2020; Oct. 25th 2020–May 9th 2021	Ideation Attempt	Diagnosis (medical record) Diagnosis (definition: a self-inflicted, potentially injurious behavior with a nonfatal outcome for which there is evidence, either explicit or implicit, of intent to die)
Kasal et al., 2022 [9]	Czechia	Repeated Cross-Sectional	General Population	Pre (3306); Peri 1(3021); Peri 2 (3000)	Pre (48.82); Peri 1 (46.84); Peri 2 (46.16)	Pre (53.7%); Peri 1 (52.3%); Peri 2 (51.1%)	Nov. 2017	May 2020; Nov. 2020	Ideation Attempt	Self-report (Items of the Mini International Neuropsychiatric Interview: Think that you would be better off dead or wish you were dead? Want to harm yourself? Think about suicide? Have a suicide plan?) Self-report (Items of Mini International Neuropsychiatric Interview: Attempt suicide? Did you ever make a suicide attempt?)
Kim et al., 2021 [12]	South Korea	Repeated Cross-Sectional	Adolescents	Pre (48,443); Peri (44,216)	Pre (15); Peri (15.1)	Pre (48.7%); Peri (47.5%)	Jun. 3rd–Jul. 12th for 2019	Aug. 3rd–Nov. 13th for 2020	Ideation Attempt	Self-report (single question: Participants were asked if they had considered suicide seriously within the past 12 months) Self-report (single question of if they had attempted suicide within the past 12 months)
King et al., 2022 [97]	Canada	Longitudinal (Matched Sample)	College Students	1330	Pre (18.5); Peri (18.8)	67.2%	2018–2019	2020–2021	Ideation Attempt	Self-report (Columbia-Suicide Severity Rating Scale)
Knudsen et al., 2021 [64]	Norway	Repeated Cross-Sectional	General Population	Pre (563); Peri 1 (691); Peri 2 (530); Peri 3 (370);	Pre (38.3); Peri 1(39.4); Peri 2 (39.2); Peri 3 (39.1)	Pre (56.6%); Peri 1(62.3%); Peri 2 (61.2%); Peri 3 (64.1%)	Jan. 28th–Mar. 11th for 2020	Mar. 12th–May 31st for 2020; Jun. 1st–Jul. 31st 2020; Aug. 1st -Sept. 18th 2020	Ideation	Diagnosis (Composite International Diagnostic Interview: Thoughts of killing oneself or wishing one was dead during the 30 days before the interview)
Koenig et al., 2021 [15]	Germany	Longitudinal (Matched Sample)	Adolescents	324	14.93	69.3%	Nov. 26th 2018–Mar. 13th 2020	Mar. 18th–Aug. 29th for 2020	Ideation Attempt	Self-report (Paykel Suicide Scale)
Kose et al., 2021 [107]	Turkey	Retrospective	Psychiatry Emergency Adolescents	Pre (128); Peri (66)	--	--	Mar.–Jun. for 2019	Mar.–Jun. for 2020	Ideation Attempt	Diagnosis (medical record)
Lee and Hong, 2022 [108]	South Korea	Repeated Cross-Sectional	High School Students	Pre (57,303); Peri 1 (54,948); Peri2 (54,848)	Pre (15.08); Peri 1 (15.19); Peri2 (15.23)	Pre (48.2%); Peri 1 (48%); Peri2 (48.1%)	Jun.–Jul. for 2019	Aug.–Nov. for 2020–2021	Ideation Attempt	Self-report (whether suicidal thoughts had occurred in the past 12 months) Self-report (whether suicide attempt had occurred in the past 12 months)
Liu et al., 2022 [61]	Canada	Repeated Cross-Sectional	General Population	Pre (57,034); Peri (5742)	Pre: 18–34 (28.4%), 35–64 (50.1%), 65+ (21.5%); Peri: 18–34 (24.8%), 35–64 (53%), 65+ (22.3%)	Pre (50.8%); Peri (50.7%)	Jan. 2nd–Dec. 24th for 2019	Feb. 1–May 7 for 2021	Ideation	Self-report (2021: have you seriously contemplated suicide since the COVID-19 pandemic began?; 2019: Have you ever seriously contemplated suicide and has this happened in the past 12 months?)
Mayne et al., 2021 [98]	United State	Repeated Cross-Sectional	Adolescents	Pre (43,504); Peri (47,684)	Pre (15.2); Peri (15.3)	Pre (49.3%); Peri (49.7%)	Jun.–Dec. for 2019	Jun.–Dec. for 2020	Ideation Attempt	Self-report (Has there been a time in the past month when you have had serious thoughts about ending your life?) Self-report (Have you ever, in your whole life, tried to kill yourself or made a suicide attempt?)
McLoughlin et al., 2022 [62]	Ireland	Retrospective	Psychiatric Emergency Adolescent Patients	Pre (15); Peri 1 (23); Peri 2 (47)	16.5	Pre (46.7%); Peri 1 (60%); Peri 2 (72%)	Mar.–May for 2019	Mar.–May for 2020–2021	Ideation	Diagnosis (medical record)
Millner et al., 2022 [109]	United State	Retrospective	Pediatric Psychiatric Patients	Pre (1096); Peri (275)	Pre (15.82); Peri (15.13)	Pre (71.1%); Peri (88.7%)	April 2017 to March 12, 2020	March 13, 2020 to April 2021	Ideation Attempt	Self-report (Self-Injurious Thoughts and Behaviors Interview Self-Report Version; past month).
Nichter et al., 2021 [56]	United State	Longitudinal	Military Veterans	3078	63.2	8.4%	Nov. 18th 2019–Mar. 8th 2020	Nov. 9th–Dec. 17th for 2020	Ideation	Self-report (Item 2 of the Suicide Behaviors Questionnaire-Revised: How often have you thought about killing yourself in the past years?)
Nsamenang et al., 2022 [110]	Canada	Retrospective	Pediatric Patients with Obesity	Pre (145); Peri (189)	Pre (11.9); Peri (11.5)	Pre (45.5%); Peri (47.1%)	Dec.15th 2019–Mar. 15th 2020	Dec.15th 2020– Mar. 15th 2021	Ideation Attempt	Diagnosis (medical record)
Reif-Leonhard et al., 2022 [113]	Germany	Retrospective	General Population	765,000	--	--	Mar.–Dec. 2019	Mar.–Dec. 2020	Attempt	Diagnosis (definition: deliberate self-harm with intend to die, irrespective of fatality probability)
Reuter et al., 2021 [99]	United State	Repeated Cross-Sectional	College Students	Ideation: Pre (619); Peri (491) Attempt: Pre (34); Peri (36)	Pre (20.3); Peri (20.6)	Pre (77.4%); Peri (83.8%)	Apr. 2018–Feb. 2020	Nov. 2020–Apr. 2021	Ideation Attempt	Self-report (During the past 12 months, did you ever seriously consider attempting suicide?) Self-report (Did you actually attempt suicide?)
S. Y. Kim et al., 2022 [74]	South Korea	Repeated Cross-Sectional	General Population	Pre (6127); Peri (5746)	Pre (51.7); Peri (52)	Pre (50.3%); Peri (50%)	Jan.–Dec. for 2019	Jan.–Dec. for 2020	Attempt	Self-report (Have you ever attempted suicide within 1 year?)
Sacco et al., 2022 [63]	United State	Retrospective	Emergency Department Patients	Pre (122,148); Peri (27,874)	--	--	Mar. 15th–Jul. 31 for 2017–2019	Mar. 15th–Jul. 31st for 2020	Ideation	Diagnosis (International Classification of Diseases, 10th edition)
Salt et al., 2022 [111]	United State	Retrospective	Adult Patients; Pediatric Patients	Pre (845,992); Peri (714,578)	--	--	Mar. 18th–Sept. 18th for 2019	Mar. 18th–Sept. 18th for 2020	Ideation Attempt	Diagnosis (International Classification of Diseases, 10th edition)
Seifert et al., 2020 [16]	Germany	Retrospective	Psychiatric Patients	Pre (476); Peri (374)	Pre (44.48); Peri (43.4)	Pre (47.9%); Peri (39.3%)	Mar. 16th–May 24th for 2019	Mar. 16th–May 24th for 2019	Ideation Attempt	Diagnosis (Arbeitsgemeinschaft für Methodik und Dokumentation in der Psychiatrie System)
Sivertsen et al., 2022 [100]	Norway	Repeated Cross-Sectional	College Students	Pre 1 (8124); Pre 2 (13,663); Pre 3 (49,836); Peri (59,028)	Pre 1 (24.1); Pre 2 (23.55); Pre 3 (24.27); Peri (23.53)	Pre 1 (65.6%); Pre 2 (65.8%); Pre 3 (66.5%); Peri (69.1%)	Oct. 11th–Nov. 8th for 2010; Feb. 24th–Mar. 27th for 2014; Feb. 6th–Apr. 5th for 2018	Mar. 1st–Apr. 6th for 2021	Ideation Attempt	Self-report (an item of the depression subscale of HSCL-25: In the past 2 weeks, including today, how much have you been bothered by thoughts of ending your life?) Self-report (an item of the Adult Psychiatric Morbidity Survey: Have you ever made an attempt to take your life, by taking an overdose of tablets or in some other way?)
Stańdo et al., 2022 [75]	Poland	Retrospective	General Population	--	13–24 (4556 million); 25–64 (20.714 million); 65+ (7.417 million)	--	2019	2020; 2021	Attempt	Diagnosis (medical record)
Taquet et al., 2022 [101]	United State	Retrospective	Eating Disorder Adolescents Patients	Pre (19,843); Peri (8471)	Pre (16.34); Peri (16.25)	Pre (75.4%); Peri (78.1%)	Jan. 20th 2017–Jan. 19th 2020	Jan. 20th 2020–Jan. 19th 2021	Ideation Attempt	Diagnosis (International Classification of Diseases, 10th edition; code R45.851) Diagnosis (International Classification of Diseases, 10th edition; code T14.91)
Thompson et al., 2021 [67]	United State	Retrospective	Psychiatric Patients	Pre (196); Peri (142)	Pre (14.53); Peri (15.06)	Pre (--); Peri (45.8%)	Apr. 13th–Sept. 14th for 2019	Apr. 13th–Sept. 14th for 2020	Attempt	Self-report (Have you made any suicide attempts in the 7 days before you came to the hospital?)
Valdez-Santiago et al., 2022 [68]	Mexico	Repeated Cross-Sectional	Adolescents	Pre (17,925); Peri (4913)	--	--	Jul. 2018–Jun. 2019	Aug.–Nov. for 2020	Attempt	Self-report (Have you ever attempted to harm yourself or deliberately cut, intoxicated or hurt yourself in any way for the purpose of dying? (2) Was this in the last 12 months?
Zhang et al., 2020 [112]	China	Longitudinal	Primary School Students	Pre (1271); Peri (1241)	12.6	40.7%	Nov. 2019	May 2020	Ideation Attempt	Self-report (Have you ever thought about killing yourself in the past 3 months?) Self-report (Have you ever tried to kill yourself in the past 3 months?)
Zhu et al., 2021 [57]	China	Longitudinal	High School Students	1393	13.04	53.1%	Sept. 2019	Jun. 2020	Ideation	Self-report (Item 9 of the Patient Heath Questionarie-9: Thoughts that you would be better off dead or of hurting yourself in some way over the past two weeks?)

--: Data not available.

**Table 2 ijerph-20-03346-t002:** Characteristics of studies reporting death by suicide.

Study	Region	Data Sources	Time Range		GDP (Peri/Pre)	Unemployment Rate (Peri/Pre)	Main Findings
			Pre-Pandemic Period	Peri-Pandemic Period			
Acharya et al., 2022 [8]	Nepal	Nepal Police Headquarter	Jul. 2017–Mar. 2020 (Jan.–Dec. 2019) *	Apr. 2020–Jun. 2021 (Apr. 2020–Mar. 2021) *	0.98	1.52	An overall increase in the monthly suicide rate was found during the pandemic months (Increase in rate = 0.28, 95% CI: 0.12, 0.45).
Appleby et al., 2021 [14]	England (NHS sustainability and transformation partnerships region)	Real Time Surveillance (RTS) System	Apr.–Oct. for 2019	Apr.–Oct. for 2020	0.98	1.14	Comparison of the suicide rates after lockdown began in 2020 to those of the same months in 2019 showed no difference (IRR = 1.00, 95% CI: 0.92–1.09).
Arya et al., 2022 [76]	India	National Crime Records Bureau (NCRB) Data	2010−2019 (2017) *	2020	1	1.48	Compared to 2017, an increase in annual suicide rate was found during the pandemic year (RR = 1.14, 95% CI: 1.13–1.14).
Chen et al., 2022 [77]	Taiwan	Taiwan’s Ministry of Health and Welfare	2017–2019	2020	1.11	1.03	Compared to previous years, a decrease in annual suicide rates after the outbreak was found (*p* = 0.05).
de la Torre-Luque et al., 2022 [78]	Spain	National Death Index	2019	2020	0.92	1.1	No significant differences in suicide mortality rates between 2019 and 2020 were found (*p* = 0.18).
Faust et al., 2021 [79]	Massachusetts, USA	Massachusetts Department of Health Registry of Vital Records and Statistics	2015–2019 (2019) *	2020	0.96	3.03	During the pandemic period, the incident rate for suicide deaths in Massachusetts was 0.67 vs. 0.80 per 100,000 person months during the corresponding period in 2019 (IRR = 0.84; 95% CI, 0.64–1.00).
Gerstner et al., 2022 [80]	Ecuador	National Directorate of Crimes against Life, Violent Deaths, Disappearances, Extortion and Kidnapping (DINASED)	Jan. 2015–Feb. 2020	Mar. 2020–Jun. 2021	0.99	1.62	During the pandemic period, suicide rate was not significantly higher than expected (RR = 0.97; 95% CI, 0.92–1.02).
Larson et al., 2022 [81]	Michigan, USA	Michigan Department of Health and Human Services (MDHHS)	Jan. 1st, 2006–Mar. 12th, 2020	Mar. 13th, 2020–Dec. 12th, 2021	0.95	2.43	Compared with before, daily suicide incidence rate declined during the pandemic for both females (9.32%; *p* = 0.01) and males (20.64%; *p* = 0.04).
McIntyre et al., 2021 [13]	Canada	Canadian National Database	Mar. 2010–Feb. 2020 (Mar. 2019–Feb. 2020) *	Mar. 2020–Feb. 2021	0.95	1.67	Overall suicide mortality rate decreased from the March 2019–February 2020 period to the March 2020–February 2021 period (absolute difference of 1300 deaths).
Mitchel and Li, 2021 [82]	Connecticut, USA	Connecticut Office of the Chief Medical Examiner	Mar. 10th–May 20th for 2014–2019	Mar. 10th–May 20th for 2020	0.96	1.64	The age-adjusted suicide rate was 13% lower than the recent 5-year average during the same period.
Page et al., 2022 [83]	Australia	Australian Bureau of Statistics	2019	2020; 2021	0.96; 1.11	1.25; 0.98	The suicide rate in 2020 was lower than the 2019 rate, while the decrease was less noteworthy when considering the trend from the beginning of 20th century.
Palacio-Mejía et al., 2022 [84]	Mexico	National Epidemiological and Statistical Subsystem of Deaths	2019	2020; 2021	0.89; 1.06	1.26; 1.26	The suicide rate for 2019, 2020, and 2021 was 4.7, 5.3, and 5.4 per 100,000 people, respectively.
Radeloff et al., 2021 [114]	Leipzig, German	Leipzig Health Authority	2010–2019	2020	1.05	0.8	In 2020, suicides rates were lower in periods with severe COVID-19 restrictions (SR = 7.2, χ^2^ = 4.033, *p* = 0.045), compared with periods without restrictions (SR = 16.8)
Reif-Leonhard et al., 2022 [113]	Frankfurt/Main, German	Institute of Legal Medicine and Communal Health Authority	Mar.–Dec. for 2019	Mar.–Dec. for 2020	0.99	1.23	The number of completed suicides did not change between March–December 2019 and March–December 2020 (IRR = 0.94, *p* > 0.05).
Rogalska and Syrkiewicz-Switała, 2022 [86]	Poland	Ministry of Health	2017–2019	2020	1.05	0.8	The total number of annual suicide attacks shows an upward trend from 2017 to 2020.
Rück et al., 2020 [87]	Sweden	Statistics Sweden	Jan.–Jun. for 2019	Jan.–Jun. for 2020	1.01	1.22	Suicide rates in January-June 2020 revealed a slight decrease compared to the corresponding rates in January–June 2019 (relative decrease by −1.2% among men and −12.8% among women).
Ryu et al., 2022 [91]	South Korea	Statistics Korea’s Microdata Integrated Service	2017–2019 (2019) *	2020	0.99	1.03	Compared to 2019 (26.9 per 100,000 people), the suicide rate declined in 2020 (25.7).
Stene-Larsen et al., 2022 [88]	Norway	Norwegian Cause of Death Registry	2010– 2019 (2019) *	2020	0.89	1.19	During the pandemic period, the observed suicide rate (12.1 per 100,000 population) was not significantly higher than expected (12.3).
Tanaka and Okamoto, 2021 [89]	Japan	Ministry of Health, Labor and Welfare	Nov. 2016–Jan. 2020	Feb.–Jun. for 2020; Jul.–Oct. for 2020	1; 0.98	1.11; 1.11	During the first 5 months of the pandemic, monthly suicide rates declined by 14%.
Wei et al., 2021 [90]	Suzhou, China	Monitoring System of Death Causes of Suzhou Center for Disease Control and Prevention	Jan.–Apr. for 2015–2019	Jan.–Apr. for 2020	1.19	0.96	Suicide was among the top five causes of death, and suicide rate had normal fluctuation during the pandemic.

CI = Confidence Interval; RR = Rate Ratio; IRR = Incidence Rate Ratio; SR = Suicide Rate. * These data were used for comparison.

**Table 3 ijerph-20-03346-t003:** Summary of meta-analysis results for studies reporting suicidal ideation in non-clinical and clinical settings.

Study Setting	No. of Studies (Samples)	Pooled PR (95% CI; *p*-Value)	Heterogeneity		
			*I* ^2^	*tau* ^2^	*Q* (*p*-Value)
Non-clinical settings	23 (28)	1.142 (1.018–1.282; *p* = 0.024)	97.734%	0.081	1191.667 (0.00)
Clinical settings	18 (23)	1.134 (1.048–1.227; *p* = 0.002)	71.029%	0.018	75.939 (0.00)

**Table 4 ijerph-20-03346-t004:** Summary of subgroup analysis for non-clinical and clinical samples reporting suicidal ideation.

Moderators	No. of Studies	Point Estimate (95% CI; *p*-Value)	Total Between		
			*Q*	df*(Q)*	*p*-Value
**Non-clinical samples**					
Population			31.838	3	0.000
Adolescents	8	1.03 (0.849–1.248; *p* = 0.296)			
Younger group ^1^	10	0.955 (0.803–1.136; *p* = 0.604)			
General population	8	2.014 (1.604–2.529; *p* = 0.000)			
Special group ^2^	2	0.833 (0.574–1.208; *p* = 0.335)			
Study design			9.941	1	0.002
Repeated cross-sectional	19	1.318 (1.132–1.535; *p* = 0.000)			
Longitudinal	8	0.842 (0.666–1.063; *p* = 0.148)			
Measurement tool			0.033	1	0.857
Self-report	25	1.139 (1.012–1.283; *p* = 0.031)			
Diagnosis	3	1.194 (0.73–1.95; *p* = 0.48)			
Timeframe for measurement			0.156	1	0.693
≤2 weeks	8	1.159 (0.95–1.414; *p* = 0.146)			
>2 weeks	19	1.217 (1.062–1.394; *p* = 0.005)			
Data collection			0.134	2	0.935
Mar–Aug 2020	16	1.141 (0.964–1.35; *p* = 0.125)			
Sept 2020–Jan 2021	8	1.118 (0.895–1.396; *p* = 0.325)			
Feb. 2021+	4	1.199 (0.884–1.628; *p* = 0.243)			
**Clinical samples**					
Population			0.944	1	0.331
Adolescent patients	15	1.171 (1.057–1.297; *p* = 0.002)			
Adult patients	8	1.081 (0.953–1.225; *p* = 0.225)			
Measurement tool			0.259	1	0.611
Self-report	2	1.079 (0.872–1.336; *p* = 0.484)			
Diagnosis	21	1.146 (1.047–1.255; *p* = 0.003)			
Timeframe for measurement			0.067	1	0.796
≤2 weeks	20	1.129 (1.129–1.027; *p* = 0.012)			
>2 weeks	3	1.158 (0.976–1.375; *p* = 0.093)			
Data collection			0.996	1	0.318
Mar–Aug 2020	19	1.12 (1.031–1.217; *p* = 0.007)			
Sept 2020–Feb 2021+ ^3^	4	1.292 (0.988–1.688; *p* = 0.061)			

^1^ An aggregate of young people (aged 19–24 years old) and college students. ^2^ An aggregate of hotline callers and military veterans. ^3^ We combined the only two samples (Irigoyen-Otiñano et al., 2022b [106]; Nsamenang et al., 2022 [110]) that collected data during Sept. 2020–Jan. 2021 and two samples after 2021 (Berger et al., 2022b [59]; McLoughlin et al., 2022b [62]).

**Table 5 ijerph-20-03346-t005:** Summary of meta-analysis results for studies reporting suicide attempt in non-clinical and clinical settings.

Study Setting	No. of Studies (Samples)	Pooled PR (95% CI; *p*-Value)	Heterogeneity		
			*I* ^2^	*tau* ^2^	*Q* (*p*-Value)
Non-clinical settings	17 (30)	1.14 (1.053–1.233; *p* = 0.001)	99.996%	0.036	7601.38 (*p* = 0.000)
Clinical settings	20 (25)	1.32 (1.17–1.489; *p* = 0.000)	70.021%	0.052	80.056 (*p* = 0.000)

**Table 6 ijerph-20-03346-t006:** Summary of subgroup analysis for non-clinical and clinical samples reporting suicide attempt.

Moderators	No. of Studies	Point Estimate (95% CI; *p*-Value)	Total Between		
			*Q*	df*(Q)*	*p*-Value
**Non-Clinical Samples**					
Population			4.005	2	0.135
Adolescent	13	1.062 (0.948–1.189; *p* = 0.3)			
Younger group ^1^	4	0.942 (0.673–1.319; *p* = 0.73)			
General population	12	1.218 (1.089–1.362; *p* = 0.001)			
Study design			2.805	2	0.246
Repeated cross-sectional	12	1.245 (1.083–1.43; *p* = 0.002)			
Longitudinal	3	1.285 (0.817–2.02; *p* = 0.277)			
Retrospective	15	1.084 (0.983–1.195; *p* = 0.106)			
Measurement tool			2.788	1	0.095
Self-report	15	1.248 (1.093–1.425; *p* = 0.001)			
Diagnosis	15	1.084 (0.983–1.196; *p* = 0.106)			
Timeframe for measurement			2.809	1	0.094
≤2 weeks	16	1.084 (0.983–1.196; *p* = 0.107)			
>2 weeks	14	1.249 (1.093–1.426; *p* = 0.001)			
Data collection			0.397	2	0.820
Mar–Aug 2020	15	1.15 (1.025–1.289; *p* = 0.017)			
Sept 2020–Jan 2021	7	1.174 (0.988–1.395; *p* = 0.069)			
Feb. 2021+	8	1.101 (0.966–1.255; *p* = 0.149)			
**Clinical samples**					
Population			1.044	1	0.307
Adolescent patients	12	1.415 (1.185–1.68; *p* = 0.000)			
Adult patients	13	1.251 (1.068–1.464; *p* = 0.005)			
Measurement tool			0.256	1	0.613
Self-report	3	1.42 (1.036–1.948; *p* = 0.13)			
Diagnosis	22	1.299 (1.132–1.492; *p* = 0.000)			
Timeframe for measurement			0.628	1	0.428
≤2 weeks	22	1.288 (1.121–1.479; *p* = 0.000)			
>2 weeks	3	1.158 (0.976–1.375; *p* = 0.012)			
Data collection					
Mar–Aug 2020	22	1.323 (1.164–1.503; *p* = 0.000)	0.016	1	0.901
Sept 2020–Feb 2021+ ^2^	3	1.288 (0.685–1.918; *p* = 0.213)			

^1^ An aggregate of young people (aged 19–24 years old) and college students. ^2^ We combined the only two samples (Irigoyen-Otiñano et al., 2022b [106]; Nsamenang et al., 2022 [110]) that collected data during Sept. 2020–Jan. 2021 and one sample after 2021 (Caballero-Bermejo et al., 2022b [70]).

## Data Availability

The data that support the findings of this study are available from the corresponding author, N.X.Y., upon reasonable request.

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
