# Peer review of "Suicide before and during the COVID-19 Pandemic: A Systematic Review with Meta-Analysis"

_ijerph, 2023, doi:10.3390/ijerph20043346_

Round 1

Reviewer 1 Report

The authors conducted a systematic review and meta-analysis of the prevalence of suicidal ideation and attempts, and rate of suicide deaths, during the COVID-19 pandemic compared to pre-pandemic. Pooling data from 72 studies (131 samples), they found a small but statistically significant increase in the prevalence of suicidal ideation and suicide attempts during the pandemic, but a non-significant decrease in the suicide death rate. Most of the effect moderators examined had small or negligible effects, however, the prevalence of suicidal ideation and attempts was significantly different between studies of adults (which showed a moderate to large increase in the prevalence of suicidal ideation and attempts) and studies of youth (which showed no increase). This study summarizes the current state of knowledge and contributes to our understanding of how suicidal thoughts and behavior have changed, or not, in the post-COVID era. It is an important analysis but would benefit from a few revisions, mostly clarifying certain methodological decisions.

Major revisions

The terms clinical and non-clinical sample need to be defined, and this should be done in the Methods section (currently it is in section 3.1 of the Results). Further, and more importantly, the authors should justify in sufficient detail their choice to split the whole analysis by this variable. The quantitative results look broadly similar between the two samples, and so it is unclear to me why they are presented separately, especially when there are more marked differences by other variables, like the age of the study samples.

More detail is needed on how the country resilience scores and government response indexes were calculated. Did the authors just take the numbers given in references 44 & 45, and if so, for what time points? Or did they calculate their own scores using the same formulas as described in the references, and if so, what were their inputs?

Minor revisions

Intro

Recommend dropping the 2nd paragraph. Most readers will know the definition of suicide and suicidal ideation, this just distracts from more relevant background.

In the 3rd paragraph, in lines 57-58 the authors state that two prior meta-analyses exist, but then in lines 66-67 they say only one does. This discrepancy should be clarified.

Methods

Search strategy: what is “the inception”? Recommend just giving the start date for the search.

Risk of bias: does a higher or lower score indicate greater risk of bias?

Conclusion

The authors should note there was no increase in suicide mortality, as this was one of the primary (and arguably the most clinically important) study outcomes.

Throughout

Note that it is no longer considered proper to say, “committed suicide.” The authors should instead say “died by suicide” or something similar.

Author Response

We appreciate that you gave us the opportunity to submit a revised manuscript entitled “Suicide Before and During the COVID-19 Pandemic: A Systematic Review with Meta-Analysis” (Manuscript ID: ijerph-2157738). We sincerely thank the time and efforts that you dedicated to providing fast and valuable feedback. We have responded to all comments and suggestions made by the editor and reviewer. In the attached Word file, our point-to-point responses are highlighted in “blue,” and changes in the revised manuscript are highlighted in “purple.

Once again, we deeply thank the editor and reviewer for the thoughtful comments and recommendations. Look forward to hearing from you.

Best Regards,

Yifei Yan,

Jianhua Hou,

Qing Li, and

Nancy Xiaonan Yu.

Reviewer 2 Report

I want to thank you for giving me the opportunity to review the article. I think that this well-designed and well-written study will make significant contributions to the literature. I congratulate the authors for this study. 

Author Response

We appreciate that you gave us the opportunity to submit a revised manuscript entitled “Suicide Before and During the COVID-19 Pandemic: A Systematic Review with Meta-Analysis” (Manuscript ID: ijerph-2157738). We sincerely thank the time and efforts that you dedicated to providing fast and valuable feedback. We have responded to the comments made by the reviewer and our response are listed below:

Reviewer #2:

I want to thank you for giving me the opportunity to review the article. I think that this well-designed and well-written study will make significant contributions to the literature. I congratulate the authors for this study.

Response:

We are very grateful for your encouraging comments. Hopefully, our study can contribute to suicide research and provide useful implications for clinical practice.

Once again, we deeply thank the editor and reviewer for the thoughtful comments and recommendations. Look forward to hearing from you.

Best Regards,

Yifei Yan,

Jianhua Hou,

Qing Li, and

Nancy Xiaonan Yu.

Reviewer 3 Report

This is a methodologically rigorous examination of a phenomenon that is consistently under reported. As expected, rates increased peripandemic and the reported confidence intervals are narrow reinforcing the reliability of the findings. The phrase "dying by suicide" is taken to be less judgmental than “committing suicide” as though suicide were a crime committed. The authors might consider changing the wording in the text.

Author Response

(The authors gave the same response as above.)

Round 2

Reviewer 1 Report

The authors have sufficiently addressed my previous concerns, and I have nothing further to add. I congratulate them on their work.